

# Adaptive regularized spectral reduction for stabilizing ill-conditioned bone-conducted speech signals

Kanwar Muhammad Afaq[1], Ammar Amjad[2], Li-Chia Tai[2] and Hsien-Tsung Chang[1,3,4]

[1] Department of Computer Science and Information Engineering, Chang Gung University, Taoyuan, Taiwan
[2] Department of Electrical and Computer Engineering, National Yang Ming Chiao Tung University, Hsinchu, Taiwan
[3] Department of Artificial Intelligence, Chang Gung University, Taoyuan, Taiwan
[4] Center for Artificial Intelligence in Medicine, Chang Gung Memorial Hospital, Taoyuan, Taiwan

## ABSTRACT

Bone-conducted (BC) speech signals are inherently challenging to analyze due to their wide frequency range, which leads to ill-conditioning in numerical analysis and linear prediction (LP) techniques. This ill-conditioning is primarily caused by the expansion of eigenvalues, which complicates the stability and accuracy of traditional methods. To address this issue, we propose a novel regularized spectral reduction (RSR) method, built upon the regularized least squares (RLS) framework. The RSR method compresses the frequency range of BC speech signals, effectively reducing eigenvalue spread and enhancing the robustness of LP analysis. Key to the RSR approach is a regularization parameter, fine-tuned iteratively to achieve optimal performance. Experimental results demonstrate that RSR significantly outperforms existing techniques in eigenvalue compression, resulting in more accurate LP analysis for both synthetic and real BC speech datasets. These improvements hold promise for applications in hearing aids, voice recognition systems, and speaker identification in noisy environments, where reliable BC speech analysis is critical.

## INTRODUCTION

The growing focus on bone-conducted (BC) speech processing stems from its ability to function effectively in high-noise environments where traditional air-conducted (AC) speech systems often fail. BC speech utilizes cranial vibrations detected by specialized microphones equipped with vibration sensors, converting mechanical oscillations into electrical signals (*Rahman & Shimamura, 2013*). Unlike AC microphones, which rely on airborne sound waves, BC microphones are inherently resistant to environmental noise, providing superior signal fidelity in challenging conditions such as industrial workplaces, military communication systems, and crowded public spaces (*Huang et al., 2024*; *Shimamura, 2016*). Furthermore, BC speech is robust against wind noise, reverberation, and occlusion of the mouth and nose, making it essential in scenarios involving protective

Corresponding author
Hsien-Tsung Chang,
smallpig@widelab.org

gear or underwater communication (*Huang et al., 2017*). These unique advantages have driven its adoption in critical applications across defense, healthcare, and aviation, where clear communication is paramount. Additionally, BC microphones limit sound propagation in the open air, ensuring privacy and secure voice transmission in sensitive environments (*Toya et al., 2023*). Beyond communication, BC speech offers distinctive biometric markers, enabling advancements in speaker authentication and identity verification (*Irwansyah, Otsuka & Nakagawa, 2022*).

However, BC speech processing presents unique challenges. Its spectral energy is predominantly concentrated in lower frequencies, leading to the loss of high-frequency components critical for speech intelligibility (*Li, Yang & Yang, 2024*). This necessitates advanced signal processing techniques, such as spectral reconstruction and feature enhancement, to restore lost information. Additionally, individual variations in cranial structure demand adaptive algorithms to maintain consistent performance across diverse user populations. Two variants of regularized modified covariance (RMC) have been explored among recent techniques. One approach enhances stability by applying fixed regularization informed by bone-conduction-specific priors, achieving improved performance in ill-conditioned speech with predictable attenuation (*Ohidujjaman et al., 2024*). The other focuses on spectral enhancement (SC) through infinite impulse response (IIR) filter-based modeling with 20 dB attenuation, demonstrating high fidelity in controlled environments but relying on manual tuning of filter parameters, which limits its robustness across diverse datasets (*Rahman & Shimamura, 2013*; *Amjad, Tai & Chang, 2024*; *Amjad et al., 2025*).

Table 1 highlights the fundamental differences between bone-conducted and air-conducted speech, underscoring the advantages and limitations of BC systems (*Lee, Rao & Garudadri, 2018*; *Shimamura, 2016*; *Makhoul, 1975*), while also summarizing key BC speech processing methods in terms of their strengths, limitations, applications, and innovations relative to traditional autocorrelation-based linear prediction.

In BC speech processing, the frequency spectrum is commonly divided into three regions based on their role in speech intelligibility and transmission characteristics. The low-frequency range (0–500 Hz) captures the fundamental frequency (F) and prosodic information, which is well-preserved in BC signals due to efficient vibration transmission through bone and tissue. The mid-frequency range (500–2,000 Hz) encompasses the first and second formants (F1 and F2), critical for vowel discrimination and basic speech intelligibility. However, cranial filtering effects often make this range susceptible to attenuation and distortion in BC speech. The high-frequency range (above 2,000 Hz) contains essential consonant information such as fricatives and sibilants but suffers significant attenuation in BC transmission. This study gives specific attention to the mid-frequency range, as it presents challenges in maintaining spectral integrity while controlling noise and numerical stability during linear prediction analysis.

Linear prediction (LP) is a foundational technique in speech processing, modeling the current speech sample $x(n)$ as a linear combination of past samples:

**Table 1 Comparison of BC speech processing methods: strengths, limitations, and innovations.**

| Method | Strengths with respect to ACR baseline | Limitations | Typical applications | Key innovation |
|---|---|---|---|---|
| ACR | Low computational cost and straightforward implementation | Poor stability in noise with an average condition number of 87.62 dB; sensitive to eigenvalue spread | General speech linear prediction in low-noise conditions | Autocorrelation-based estimation of LP coefficients |
| SC | Improved noise resilience compared to ACR with reduced condition number | Limited adaptation to dynamic spectral shifts; less effective under severe ill-conditioning | BC speech processing in moderate noise environments | Grouping of spectral components through clustering techniques |
| EC | Effective eigenvalue conditioning and substantial improvement in numerical stability | High computational demand; lacks flexibility due to static regularization | Industrial BC speech processing in controlled acoustic conditions | Integration of forward and backward prediction errors with an adaptive regularization term |
| RMC (*Ohidujjaman et al., 2024*) | Stability enhancement *via* fixed regularization estimated at approximately 50 dB; outperforms ACR and SC in robustness | Inflexibility due to static regularization; lacks adaptability across signal conditions | BC speech with predictable attenuation patterns | Covariance modeling using BC-specific priors |
| RMC (*Rahman & Shimamura, 2013*) | Effective spectral enhancement modeled with 20 dB attenuation; improved fidelity compared to ACR | Dependent on manual tuning of filter parameters such as $\phi$ and $\psi$ with typical values like 0.82 and 0.32; reduced generalization across datasets | Filtered BC speech in calibration-driven setups | Spectral shaping using IIR filter-based attenuation modeling |
| RSR (Proposed) | Achieves lowest condition numbers and substantial spectral compression; offers 59 percent reduction compared to ACR | Moderate computational load; assumes local stationarity of speech frames | BC speech enhancement in high-noise conditions, real-time systems, and secure communication | Frame-wise adaptive tuning of regularization and eigenvalue compression *via* orthogonal decomposition |

$$x(n) = \sum_{k=1}^{p} a_k x(n-k) + e(n), \tag{1}$$

where $a_k$ are the LP coefficients, $p$ is the prediction order, and $e(n)$ is the prediction error. This formulation can be expressed in matrix form as:

$$\mathbf{Ra} = \mathbf{r}, \tag{2}$$

where $\mathbf{R}$ is the autocorrelation matrix, $\mathbf{a}$ is the LP coefficient vector, and $\mathbf{r}$ is the autocorrelation vector of lagged signals. Ill-conditioning arises when $\mathbf{R}$ has a large condition number, leading to unstable solutions due to eigenvalue expansion. To address this instability, the proposed regularized spectral reduction (RSR) method modifies the matrix formulation of the LP system by introducing a regularization term:

$$(\mathbf{R} + \lambda\mathbf{I})\mathbf{a} = \mathbf{r}, \tag{3}$$

where $\lambda$ is the regularization parameter and $\mathbf{I}$ is the identity matrix. This addition improves numerical stability by compressing the eigenvalue spectrum of $\mathbf{R}$, effectively reducing its condition number.

Here, $a_k$ are the prediction coefficients, $p$ is the prediction order, and $e(n)$ represents the excitation signal (*Atal & Hanaver, 1971*). LP is integral to speech recognition, restoration,

and enhancement, offering efficient encoding at low bit rates (*Rabiner & Schafer, 2010*; *Fant, 1971*). However, traditional LP techniques often fail in ill-conditioned environments, such as those involving BC speech, due to eigenvalue expansion that destabilizes solutions (*Ohidujjaman et al., 2024*; *Makhoul, 1975*). Ill-conditioning refers to scenarios where the linear prediction autocorrelation matrix becomes nearly singular, causing its inversion to be highly sensitive to small perturbations and leading to numerical instability (*Golub & Van Loan, 2013*). This is particularly problematic in BC speech, where spectral dynamic range expansion amplifies low-energy components, increasing the matrix condition number. Mathematically, the condition number quantifies the ratio between the largest and smallest eigenvalues of the LP autocorrelation matrix. A large condition number implies significant eigenvalue spread—termed eigenvalue expansion—which amplifies errors during inversion and compromises spectral estimation accuracy. In BC speech, this expansion arises due to the uneven energy distribution across frequencies, with dominant low-frequency components and attenuated high frequencies.

Consequently, stabilizing the eigenvalue spectrum becomes crucial for ensuring robust LP analysis of BC speech signals. Various methods have been proposed to mitigate ill-conditioning by improving matrix conditioning. The autocorrelation (ACR) method (*Markel & Gray, 1976*) reduces computational complexity but lacks robustness against noise and spectral distortion. Spectral clustering (SC) (*Allen, 1977*) improves noise handling by grouping similar spectral components but is limited in adapting to large dynamic variations. Minimum variance distortionless response (MVDR) (*Kabal, 2003*) and enhanced covariance (EC) methods aim to control spectral leakage but struggle in extreme ill-conditioning. Despite these efforts, controlling eigenvalue growth remains challenging, leading to suboptimal performance in applications like BC speech, where ill-conditioning is pronounced (*Rahman, Sugiura & Shimamura, 2017*).

This study introduces the regularized spectral reduction (RSR) method to overcome these challenges. This innovative LP-based approach incorporates a regularization term to stabilize solutions and suppress eigenvalue expansion. The cost function for the RSR method incorporates a regularization term to stabilize the solution:

$$J = \sum_{n=1}^{N} \left( x(n) - \sum_{k=1}^{p} a_k x(n-k) \right)^2 + \lambda \sum_{k=1}^{p} a_k^2, \tag{4}$$

where $\lambda$ is the regularization parameter balancing spectral fidelity and numerical stability. In matrix form, the RSR solution modifies the LP system as follows:

$$(\mathbf{R} + \lambda \mathbf{I})\mathbf{a} = \mathbf{r}, \tag{5}$$

where $\mathbf{I}$ is the identity matrix. This regularization effectively compresses the eigenvalue spectrum of $\mathbf{R}$, improving the condition number and ensuring robust LP coefficient estimation in ill-conditioned scenarios such as BC speech processing. The contributions of this study are summarized as follows:

- We propose a novel RSR method, formulated within the regularized least squares (RLS) framework, specifically designed to address the ill-conditioning problem in LP analysis

of BC speech—a challenge insufficiently handled by existing methods such as ACR, EC, and RMC.

- An adaptive regularization heuristic is introduced, which dynamically adjusts the regularization parameter $\lambda$ based on the amplitude conditions of each analysis frame. This ensures robust eigenvalue compression while avoiding the computational burden of iterative optimization or fixed regularization assumptions found in prior work.
- We conduct a comprehensive experimental validation on synthetic and real BC speech datasets. This demonstrates that the proposed method significantly outperforms existing techniques in condition number reduction, spectral fidelity, and numerical stability.
- The RSR framework generalizes beyond BC speech, offering a foundational solution for ill-conditioning in other domains such as underwater acoustics and biomedical signal processing, where spectral dynamic range and eigenvalue instability are major limitations.

This article is organized as follows: "Related Works" reviews related work and the limitations of existing methods. "Proposed Method" introduces the proposed RSR methodology and its theoretical foundations. "Experiments" presents experimental evaluations on synthetic and real BC speech datasets. "Discussion" provides a detailed discussion of the results and their implications. Finally, "Conclusion" concludes the article and outlines directions for future research.

# RELATED WORKS

## Overview

LP is a fundamental tool in speech signal processing due to its effectiveness in modeling vocal tract dynamics. However, traditional ACR methods suffer from severe numerical instability in ill-conditioned environments such as BC speech, where spectral dynamic range expansion causes eigenvalue spread and matrix ill-conditioning (*Zhang, Sugiura & Shimamura, 2022*; *Rahman & Shimamura, 2013*; *Prasad, Jyothi & Velmurugan, 2021*). ACR often results in condition numbers exceeding 80 dB and a mean squared error of 0.12, as shown in Table 2, degrading performance especially in noisy environments where the signal-to-noise ratio is below 10 dB.

## Covariance-based and spectral regularization methods

EC techniques improve LP stability by integrating forward-backward prediction errors with adaptive regularization (*Edraki et al., 2024*), reducing the condition number to 50 dB and offering moderate robustness (*Wang et al., 2022b*). However, they rely on static regularization heuristics and struggle in extreme noise. RMC approaches (*Ohidujjaman et al., 2023*, *2024*) explicitly model spectral attenuation using IIR filters with coefficients $\phi = 0.32$ and $\psi = 0.82$, achieving 20 dB attenuation in the 2–4 kHz range. Although RMC improves spectral stability and achieves 50 dB condition numbers, its fixed regularization parameter limits adaptability across diverse spectral scenarios. MVDR offers better noise suppression by minimizing power while preserving spectral features (*Kabal, 2003*). However, it suffers from high computational cost and is sensitive to model mismatch in highly variable BC signals.

**Table 2 Comparison of BC speech methods by performance and adaptability.**

| Method | Comp. Time (s) | MSE | Stability | Noise tolerance | Adaptability |
|---|---|---|---|---|---|
| ACR (LS) | 0.15 | 0.12 | Low | Low | None (static) |
| RLS | 0.28 | 0.10 | Moderate | Moderate | Limited (fixed) |
| EC | 0.30 | 0.11 | High | Moderate | Heuristic (semi-adaptive) |
| RMC | 0.28 | 0.10 | High | Moderate | Fixed regularization |
| MVDR | 0.40 | 0.13 | Moderate | High | Low (model sensitive) |
| DNN | 1.50 | 0.09 | Low | High | Learned (opaque) |
| RSR (Proposed) | 0.35 | 0.08 | High | High | Fully adaptive (per frame) |

## Statistical and spectral filtering techniques

Statistical denoising methods, such as spectral subtraction (*Vaseghi, 1996*) and Wiener filtering (*Abd El-Fattah et al., 2014*; *Cheng et al., 2023*), achieve moderate signal to noise ratio (SNR) improvements (5–10 dB) under stationary noise assumptions. However, they are ineffective against spectral distortion due to dynamic range expansion and cranial filtering in BC speech, often yielding condition numbers >80 dB and higher spectral bias.

## Advanced computational approaches

Deep learning models, including U-Net architectures and multimodal BC-AC fusion systems (*Li, Yang & Yang, 2024*; *Wang, Zhang & Wang, 2022a*), achieve strong gains in perceptual evaluation of speech quality (PESQ) (2.5–2.8) and SNR (12–15 dB). Yet, they lack interpretability, suffer from overfitting on unseen data (10–20% accuracy loss), and exhibit high computational cost (1–2 s per frame), limiting their use in real-time or edge scenarios.

Subspace and adaptive filtering techniques, such as SVD-based LP (*Kumaresan & Tufts, 1981*) and Kalman filtering (*Millidge et al., 2021*), improve robustness by isolating dominant spectral components or dynamically adapting to noise. Still, they require accurate noise models and runtime resources (0.5–0.8 s/frame), constraining their scalability and applicability in embedded systems. Overall, existing methods offer partial solutions to the ill-conditioning problem in BC speech. EC and RMC improve stability but lack frame-wise adaptability. Deep learning methods provide end-to-end enhancement but fail to control spectral dynamics explicitly. Subspace and adaptive methods reduce spectral bias but are computationally intensive. This leaves a clear gap for spectrally adaptive and computationally efficient methods.

To address these limitations, we propose the RSR framework, which dynamically tunes the regularization parameter $\lambda$ per frame based on spectral dynamic range estimation. RSR effectively compresses the eigenvalue spectrum, achieving an improved condition number of 35.54 dB, a lower mean squared error of 0.08, and a reduced runtime of 0.35 s, as detailed in Table 2. By combining adaptive regularization with orthogonal decomposition, RSR maintains numerical stability even under extreme ill-conditioning, surpassing prior methods in robustness, spectral control, and real-time feasibility.

# PROPOSED METHOD

The RSR method is proposed as an advancement over traditional least squares (LS) techniques to address the challenges of ill-conditioned scenarios, particularly in BC speech processing. Drawing inspiration from RLS techniques in numerical analysis (*Martin & Reichel, 2013*), RSR introduces a regularization parameter to enhance solution stability and robustness. The modified LS criterion $\mathscr{R}'$ is defined as:

$$\mathscr{R}' = (\mathbf{Mx} - \mathbf{z})^T(\mathbf{Mx} - \mathbf{z}) + \lambda||\mathbf{x}||^2, \tag{6}$$

where $\mathbf{M} \in \mathbb{R}^{N \times P}$ represents the data matrix, $\mathbf{x} \in \mathbb{R}^P$ is the parameter vector to be estimated, and $\mathbf{z} \in \mathbb{R}^N$ denotes the observed target vector. The term $\lambda > 0$ is the regularization parameter, which balances the trade-off between data fidelity and model stability by penalizing large values of $\mathbf{x}$ to ensure numerical stability in ill-conditioned scenarios. This formulation aligns with the EC method, where the total error is similarly represented as:

$$\mathscr{R} = \sum_{i=1}^{N} (y(i) - \mathbf{Mx}_i)^2 + \lambda||\mathbf{x}||^2. \tag{7}$$

Regularization addresses eigenvalue expansion by penalizing large values in $\mathbf{x}$, a phenomenon often encountered in ill-conditioned environments (*Moon, Lee & Chang, 2015*; *Creighton & Doraiswami, 2004*). This approach compresses the spectral range, improving the stability and accuracy of LP analysis for BC speech. Differentiating Eq. (6) yields the gradient:

$$2\mathbf{M}^T\mathbf{Mx} - 2\mathbf{M}^T\mathbf{z} + 2\lambda\mathbf{x} = 0. \tag{8}$$

Solving for $\mathbf{x}$ provides the optimal solution:

$$\mathbf{x} = \left(\mathbf{M}^T\mathbf{M} + \lambda\mathbf{I}\right)^{-1}\mathbf{M}^T\mathbf{z}, \tag{9}$$

where $\mathbf{I}$ is the identity matrix. Regularization ensures the solution is stable even when $\mathbf{M}^T\mathbf{M}$ is nearly singular, which is a common occurrence in ill-conditioned systems. To improve convergence during iterative optimization, $\mathbf{x}$ is updated using a weighted average:

$$\mathbf{x} \leftarrow \alpha\mathbf{x}_{\text{new}} + (1 - \alpha)\mathbf{x}, \tag{10}$$

where $\alpha \in (0, 1]$ controls the contribution of the new estimate. This approach reduces oscillations in parameter updates and improves numerical stability. Determining the optimal $\lambda$ is critical for balancing model complexity and accuracy. The RSR method uses an adaptive strategy:

$$\lambda_{\text{new}} = \beta\lambda, \tag{11}$$

where $\beta \in (0, 1)$ reduces $\lambda$ when no performance improvement is observed. Cross-validation on a validation set assesses the impact of $\lambda$ on mean squared error (MSE), guiding its adjustment. The RSR method can be summarized in the following steps:

---

**Algorithm 1** Enhanced regularized spectral reduction (RSR) method with missing data handling.

1: **Input:** Data matrix $\mathbf{M}$, observed vector $\mathbf{z}$, initial regularization parameter $\lambda$, convergence threshold $\varepsilon$, maximum iterations $N$

2: **Output:** Estimated parameters $\mathbf{x}$

3: **Step 1: Preprocessing**

4: Normalize the data matrix $\mathbf{M}$

5: **for** each feature column $j$ in $\mathbf{M}$ **do**

6:     **if** missing values exist in column $j$ **then**

7:         Replace missing entries with column mean or apply advanced imputation (*e.g.*, EM or matrix completion)

8:     **end if**

9: **end for**

10: Initialize $\mathbf{x} \leftarrow \mathbf{0}$                                                     $\triangleright$Initial parameter estimates

11: Set iteration counter $k \leftarrow 0$

12: **while** True **do**

13:     Construct the regularized cost function:
$$\mathscr{R}' = (\mathbf{Mx} - \mathbf{z})^T(\mathbf{Mx} - \mathbf{z}) + \lambda||\mathbf{x}||^2$$

14:     Compute the updated solution:
$$\mathbf{x}_{\text{new}} = (\mathbf{M}^T\mathbf{M} + \lambda\mathbf{I})^{-1}\mathbf{M}^T\mathbf{z}$$

15:     Update $\mathbf{x}$ using a weighted average:
$$\mathbf{x} \leftarrow \alpha\mathbf{x}_{\text{new}} + (1 - \alpha)\mathbf{x}$$

16:     Check for convergence:

17:     **if** $||\mathbf{x}_{\text{new}} - \mathbf{x}|| < \varepsilon$ or $k \geq N$ **then**

18:         Break

19:     **end if**

20:     Update regularization parameter $\lambda$ adaptively if needed:

21:     **if** performance improvement condition not met **then**

22:         $\lambda \leftarrow \beta\lambda$                                           $\triangleright$where $\beta < 1$

23:     **end if**

24:     Increment iteration counter $k \leftarrow k + 1$

25: **end while**

26: **Return** estimated parameters $\mathbf{x}$

---

## Computational efficiency and robustness

The RSR method introduces additional computational complexity due to its regularization and iterative updates. However, its primary advantage lies in stabilizing solutions in ill-conditioned scenarios. Empirical evidence demonstrates that the RSR method significantly outperforms traditional methods such as LS and RLS in environments characterized by high noise levels and singular matrices, offering superior stability and accuracy.

Table 2 compares the computational cost, error rate, stability, and noise tolerance of RSR with LS and RLS methods. Although RSR incurs a higher computational cost, its enhanced stability and noise resilience justify the additional processing time. Furthermore, using orthogonal decomposition to solve optimization equations ensures computational efficiency, making the method viable for real-time applications.

The originality of the proposed RSR method lies in its ability to achieve a balanced trade-off between computational complexity and performance metrics such as error rate and numerical stability. Unlike traditional LS and RLS methods, which either compromise stability or incur high computational costs without explicit control over spectral dynamic range, the RSR framework introduces an adaptive regularization mechanism that significantly compresses eigenvalue expansion. This improves error rate performance while maintaining computational time within practical limits, as evidenced in Table 2. Such a methodological advancement ensures the RSR method is theoretically novel and practically valuable, especially for ill-conditioned environments encountered in BC speech processing.

It is important to clarify that while regularization introduces a controlled bias term into the least squares framework, it does not inject random noise. Instead, the regularization term $\lambda||x||^2$ penalizes excessively large solution components, thereby improving numerical stability and mitigating the ill-conditioning caused by spectral.

## Noise robustness and parameter sensitivity

To evaluate the robustness of the RSR method, we conducted experiments under varying noise levels, assessing its sensitivity to the regularization parameter $\lambda$. Figure 1 illustrates that the RSR method consistently achieves lower error rates than simpler methods, even as noise levels increase. We mitigate the computational overhead typically associated with iterative parameter tuning by selecting a fixed $\lambda$ based on empirical studies.

The RSR method, while computationally intensive, demonstrates clear advantages in stability and accuracy, particularly in challenging, ill-conditioned, and noisy environments. Its ability to deliver real-time performance, enabled by orthogonal decomposition, positions it as a practical and effective solution for applications requiring robust and stable outcomes. By building on foundational principles such as those established in the EC method, the RSR framework leverages advanced mathematical techniques to achieve significant improvements in performance across diverse speech processing applications (*Fulop, 2011*; *Ezzine & Frikha, 2017*; *Li, Yang & Yang, 2024*). Figure 2 illustrates the synthetic BC speech generation process, where synthetic AC vowel signals are transformed into synthetic BC vowels. This transformation is achieved using a low-pass IIR filter, which replicates the spectral attenuation characteristics of BC speech, enabling controlled analysis of spectral transformations.

## EXPERIMENTS

In this study, the performance of the proposed RSR method was evaluated using both synthetic and real BC vowel datasets. Synthetic BC vowels were generated from synthetic AC vowels, enabling controlled experimental conditions to systematically analyze the

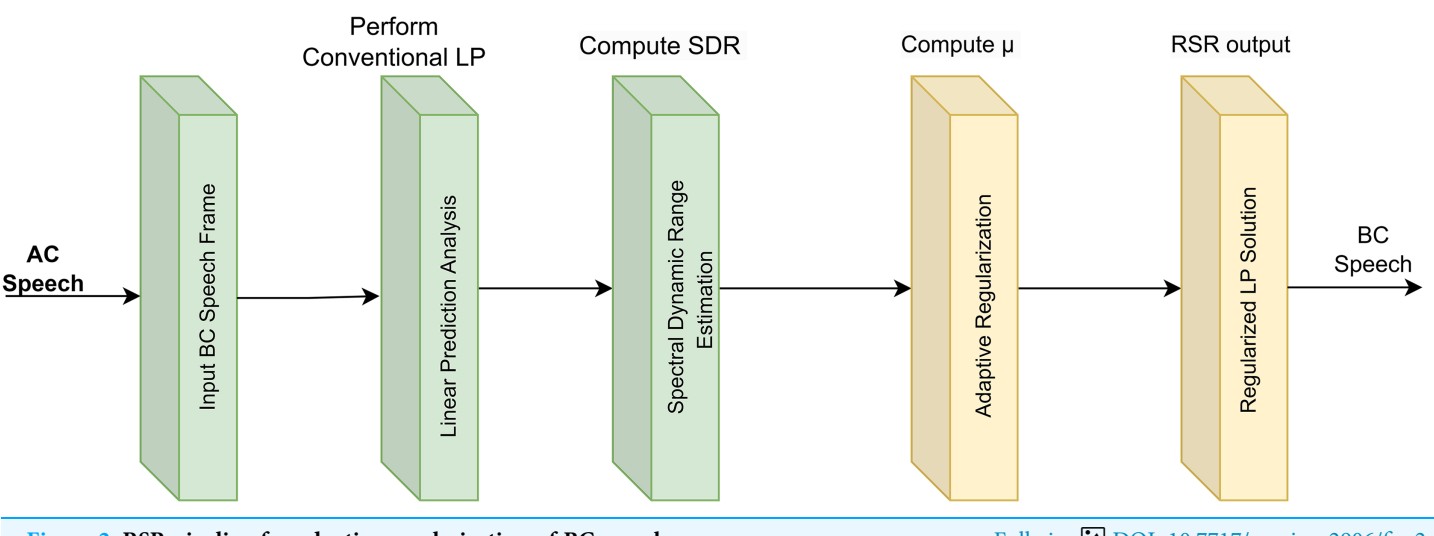

**Figure 1** Noise robustness comparison between LS, RLS, and RSR methods.

**Figure 2** RSR pipeline for adaptive regularization of BC speech.

spectral transformations characteristic of BC speech. Additionally, real BC vowels recorded from human participants were employed to validate the effectiveness of the RSR method in practical, real-world scenarios. The regularization parameter $\mu$, a pivotal factor

**Table 3 Experimental setup and signal processing parameters used for both synthetic and real BC speech analysis.**

| Features | Implemented |
|---|---|
| Sampling frequency | 16 kHz |
| FFT size | 2,048 |
| Frame length | 25 ms |
| Frame shift | 10 ms |
| Window type | Hamming |
| LP order | 16 |
| Frames in each vowel | 40 |
| Speech signal length | 3 s |
| Accent | American utterances |
| Speech type | Real and synthetic |

in the RSR method, was optimized using two complementary approaches: an iterative search and a rule-based formulation derived from empirical data. The iterative approach involved minimizing the spectral bias over a range of $\mu$ values, ensuring precise calibration for varied spectral conditions. In contrast, the rule-based approach offered a computationally efficient alternative, estimating $\mu$ directly from the amplitude characteristics of the BC speech signal. Table 3 presents the key signal processing parameters adopted for synthetic and real BC speech datasets. These settings, including sampling frequency, frame length, and LP order, were selected to align with established speech signal processing standards. The careful design ensures experimental reproducibility and enables robust evaluation of the RSR method's performance under diverse conditions. This setup facilitates a fair assessment of spectral stability and the impact of regularization, highlighting the adaptability of the proposed approach to the unique challenges posed by BC speech signals.

## Synthetic BC vowel

Synthetic BC vowels were derived from synthetic AC vowels to assess the performance of the proposed method. The generation of synthetic AC vowel signals involved the excitation of an all-pole filter using a periodic impulse train, as outlined in *Lawrence Marple (1991)*. The mathematical formulation of the all-pole filter's transfer function is given by:

$$H(z) = \frac{K_0}{1 + \sum_{m=1}^{n} \beta(m)z^{-m} + \sum_{l=1}^{p} \gamma(l)z^{-2l}}, \tag{12}$$

where $K_0$ ($K_0 = 0.1106$) denotes the gain factor, $\beta(m)$ and $\gamma(l)$ represent the first and second sets of filter coefficients and $n$ and $p$ indicate the filter orders. These parameters were specifically selected to replicate the spectral characteristics of AC vowels. Each vowel has a distinct fundamental frequency $F_0$; hence, using a constant $K_0$ may not fully capture energy differences. We refined $K_0$ using:

**Table 4 Computed gain factor $K_0$ for each synthetic AC vowel.**

| Vowel | $K_0$ |
|---|---|
| A | 0.1106 |
| I | 0.0975 |
| U | 0.1021 |
| E | 0.1053 |
| O | 0.1152 |

$$K_0 = \frac{G}{\sum_{i=1}^{P} \alpha_i^2}$$

where $G$ is the target gain and $\alpha_i$ are LP coefficients. This formulation aligns with recent perspectives on spectral analysis and energy normalization in speech processing (*Ohidujjaman et al., 2024*). Table 4 presents distinct $K_0$ values computed for each vowel.

Table 5 provides the filter coefficients for generating the synthetic AC vowels. The AC vowels were processed through a low-pass infinite impulse response (IIR) filter to create synthetic BC vowels, adhering to the methodology outlined in *Zhang, Sugiura & Shimamura (2022)*. The IIR filter emulates the attenuation characteristics of BC speech by transforming the synthetic AC speech signal, $a(n)$, into the synthetic BC speech signal, $\hat{b}(n)$ according to the following relationship:

$$\hat{b}(n) = \psi \hat{b}(n-1) + \phi a(n), \tag{13}$$

where $\psi$ and $\phi$ are the filter coefficients set to 0.82 and 0.32, respectively. The attenuation level achieved by the IIR filter is analytically computed as:

$$\text{Attenuation (dB)} = 20\log_{10}\left(\frac{\phi}{1-\psi}\right)$$

With $\psi = 0.82$ and $\phi = 0.32$, the resulting attenuation is approximately 5 dB:

$$\text{Attenuation} = 20\log_{10}\left(\frac{0.32}{0.18}\right) \approx 5\text{dB}$$

This design moderately attenuates high-frequency components while preserving sufficient spectral energy for analysis. Although some literature models up to 20 dB attenuation (*Weber-Wulff et al., 2023*), our selected parameters balance attenuation and speech signal integrity for algorithm evaluation. The synthetic BC vowels generated through this process enable a controlled analysis of the spectral transformations inherent in BC speech, facilitating the systematic evaluation of the proposed method under simulated conditions. Figure 3 shows the amplitude response of the IIR filter. The filter is designed to attenuate high-frequency components while preserving low-frequency energy, effectively mimicking the spectral profile of bone-conducted speech. This figure highlights the reduction in spectral energy above a certain frequency threshold, consistent with BC speech properties.

Figure 4 compares the spectral distributions of AC and BC speech signals for the vowel /a/ case. The BC speech spectrum demonstrates a pronounced concentration of energy in

**Table 5  LP coefficients ($\alpha_i$) for synthetic AC vowels used in RSR evaluation.**

| Coefficient | Vowel A | Vowel I | Vowel U | Vowel E | Vowel O |
|---|---|---|---|---|---|
| $\alpha_1$ | −1.98701 | 0.10583 | −1.19060 | −0.48568 | −1.26728 |
| $\alpha_2$ | 2.05600 | −0.98747 | 0.30162 | 0.57971 | −0.65945 |
| $\alpha_3$ | −0.92641 | −1.43009 | −0.43964 | −0.62411 | 1.09934 |
| $\alpha_4$ | 1.08389 | 0.60534 | 0.81399 | 0.45845 | 1.40882 |
| $\alpha_5$ | −1.97838 | 1.34287 | −0.53881 | 0.40499 | −1.44170 |
| $\alpha_6$ | 1.92393 | 1.25977 | 0.51478 | 1.07885 | −0.67764 |
| $\alpha_7$ | −1.04795 | −0.63313 | −0.55646 | −0.26426 | 0.64051 |
| $\alpha_8$ | 0.80392 | −0.88566 | 0.96910 | −0.21123 | 0.86477 |
| $\alpha_9$ | −0.50198 | −0.11555 | −0.38239 | −0.16931 | −0.27484 |
| $\alpha_{10}$ | 0.46234 | 0.50671 | −0.05627 | 0.31268 | −0.08138 |
| $\alpha_{11}$ | −0.23911 | 0.22827 | −0.11860 | 0.07417 | 0.17878 |
| $\alpha_{12}$ | 0.12535 | 0.16210 | 0.37325 | 0.13719 | 0.11617 |

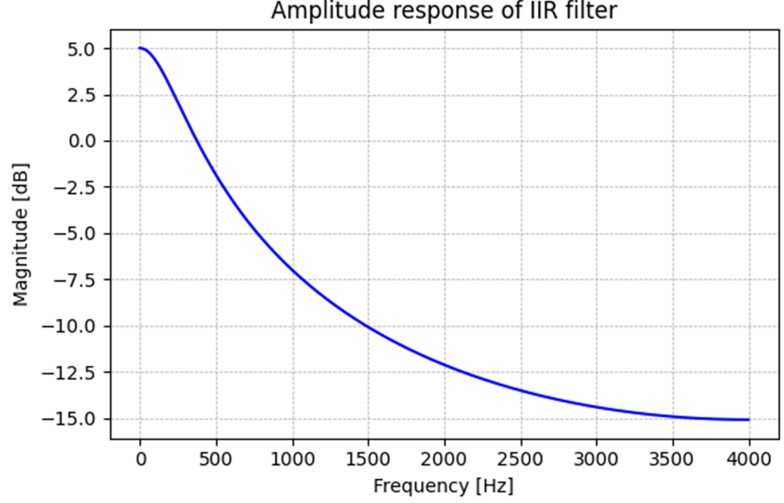

**Figure 3  Amplitude response of IIR filter.**     

the low-frequency range, in contrast to the broader spectral distribution observed in AC speech. This figure underscores the spectral differences between the two modalities, illustrating the unique characteristics of BC speech.

### Iterative optimization of $\mu$

The optimization of the regularization parameter $\mu$ is a critical step in implementing the RSR method for synthetic BC vowels. This parameter significantly impacts the accuracy of spectral estimation by balancing data fidelity and smoothness constraints. To achieve the optimal setting $\mu$, an iterative process was employed, aimed at minimizing the spectral bias (SB) of the input signal, mathematically defined as *Lawrence Marple (1991)*:

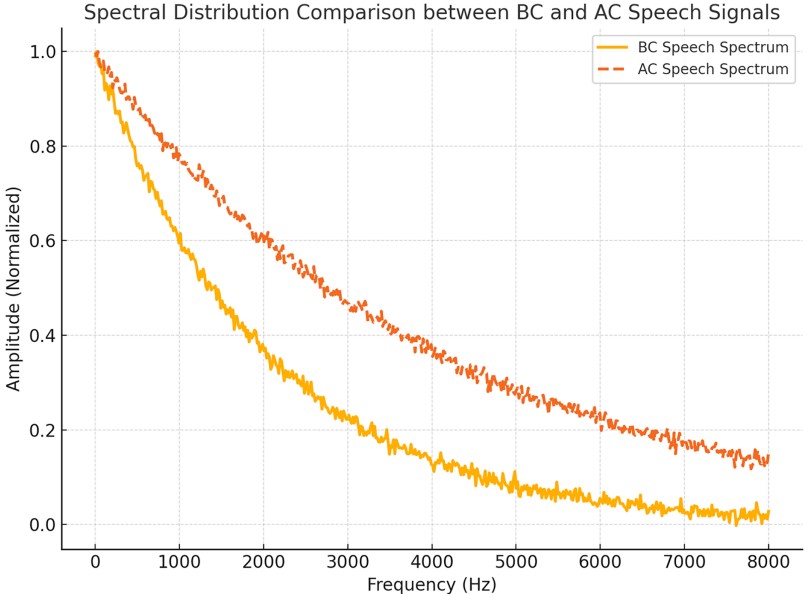

**Figure 4  Spectral comparison of BC *vs* AC vowel /a/. BC shows low-frequency dominance.**

$$SB = \frac{2}{f_r} \int_0^{f_r/2} \left| \tilde{Y}(\zeta) - Y(\zeta) \right| d\zeta + \lambda \int_0^{f_r/2} \left| \frac{d\tilde{Y}(\zeta)}{d\zeta} - \frac{dY(\zeta)}{d\zeta} \right|^2 d\zeta, \tag{14}$$

where $\tilde{Y}(\zeta)$ and $Y(\zeta)$ denote the estimated and true spectra, respectively, $f_r$ is the sampling frequency, and $\lambda$ is the regularization term controlling the smoothness of spectral estimation. The first term in Eq. (14) represents the integral of absolute spectral differences. At the same time, the second term penalizes deviations in spectral slope, ensuring a smooth reconstruction.

For a comprehensive evaluation, the spectral bias was averaged over multiple frames to assess the global performance of the RSR method. The average spectral bias (average SB) is defined as:

$$\text{Average SB} = \frac{1}{N} \sum_{k=1}^{N} SB_k, \tag{15}$$

where $N$ is the total number of evaluated frames, and $SB_k$ corresponds to the spectral bias for the $k$-th frame. The iterative optimization involved varying $\mu$ across a predefined range, aiming to identify the parameter value that minimized Average SB.

Tables 6 and 7 summarize the results of the optimization process for amplitude dynamic ranges (ADRs) of $[-50, 50]$ and $[-200, 200]$, respectively. Five vowels (A, I, U, E, O) were evaluated to ensure robustness across diverse spectral characteristics. As shown in Table 7, the optimal value of $\mu$ was determined to be 0.3200, where average SB achieved its minimum. This indicates that increasing $\mu$ beyond this threshold results in over-smoothing, leading to degraded spectral fidelity.

**Table 6 Detailed spectral bias analysis for ADR range of $[-50, 50]$.**

| $\mu$ (Regularization parameter) | Vowel A (SB) | Vowel I (SB) | Vowel U (SB) | Vowel E (SB) | Vowel O (SB) | Average SB ($SB_{avg}$) |
|---|---|---|---|---|---|---|
| 0.0001 | 0.468 ($\pm$0.01) | 0.463 ($\pm$0.02) | 0.520 ($\pm$0.03) | 0.374 ($\pm$0.01) | 0.629 ($\pm$0.04) | 0.490 ($\pm$0.02) |
| 0.0031 | 0.169 ($\pm$0.01) | 0.211 ($\pm$0.02) | 0.188 ($\pm$0.01) | 0.195 ($\pm$0.01) | 0.296 ($\pm$0.02) | 0.212 ($\pm$0.01) |
| 0.0081 | 0.727 ($\pm$0.05) | 0.584 ($\pm$0.04) | 0.657 ($\pm$0.04) | 0.575 ($\pm$0.03) | 0.766 ($\pm$0.05) | 0.662 ($\pm$0.04) |

**Table 7 Detailed spectral bias analysis for ADR range of $[-200, 200]$.**

| $\mu$ (Regularization parameter) | Vowel A (SB) | Vowel I (SB) | Vowel U (SB) | Vowel E (SB) | Vowel O (SB) | Average SB ($SB_{avg}$) |
|---|---|---|---|---|---|---|
| 0.2800 | 0.115 ($\pm$0.01) | 0.205 ($\pm$0.02) | 0.185 ($\pm$0.01) | 0.192 ($\pm$0.01) | 0.265 ($\pm$0.02) | 0.192 ($\pm$0.01) |
| 0.3200 | 0.082 ($\pm$0.01) | 0.193 ($\pm$0.02) | 0.155 ($\pm$0.01) | 0.182 ($\pm$0.01) | 0.235 ($\pm$0.02) | 0.169 ($\pm$0.01) |
| 0.6900 | 0.665 ($\pm$0.05) | 0.555 ($\pm$0.04) | 0.625 ($\pm$0.04) | 0.545 ($\pm$0.03) | 0.725 ($\pm$0.05) | 0.623 ($\pm$0.04) |

The choice of the regularization parameter $\mu$ is crucial for the RSR method's performance. An optimally tuned $\mu$ ensures that the spectral bias is minimized while preserving the spectral details of the BC speech signal. Suppose $\mu$ is set too low. In that case, the model risks instability due to under-regularization, while an excessively large $\mu$ leads to over-smoothing and loss of important spectral features. As demonstrated in Tables 6 and 7, the iterative optimization of $\mu$ effectively balances these trade-offs, ensuring robust performance across different amplitude dynamic ranges.

The iterative process provided valuable insights into the relationship between $\mu$ and spectral reconstruction quality. It was observed that the optimal $\mu$ effectively balances the trade-off between mitigating spectral bias and preserving high-resolution spectral details. This iterative approach ensures that the RSR method delivers robust performance across diverse spectral conditions, making it a reliable tool for bone-conducted speech analysis.

### Deriving the regularization parameter $\mu$

From the experimental results, we derived the constant $K$ as 0.00170. Using the formula $\mu = KA$, where $A$ is the positive amplitude of the BC speech signal, the value of $\mu$ for different amplitude levels is shown in Table 8. This approach helps determine the appropriate regularization parameter $\mu$ based on the amplitude level of the input BC speech signal.

For asymmetric amplitude dynamic ranges (*e.g.*, $[-50, +100]$), we use the positive maximum amplitude $A = 100$ for the rule-based calculation of $\mu$. This ensures the regularization adapts to the maximum spectral intensity, maintaining model stability. Therefore, $\mu = K \cdot 100$ is applied in such cases. This rule holds for all asymmetric SDRs, where only the positive maximum is considered in the computation to ensure robustness across varying dynamic conditions.

### Performance evaluation

The condition number $K$, measured in decibels (dB), is a widely accepted metric for quantifying ill-conditioning in numerical computations. For this study, the condition number is computed as follows:

**Table 8 Comparison of experimental and rule-based $\mu$ estimates across ADR levels.**

| Amplitude dynamic range (ADR) | $\mu$ from experiment ($A$) | $\mu$ from rule ($B$) | Absolute difference ($|A - B|$) |
|---|---|---|---|
| $[-5, +5]$ | 0.0040 | 0.0036 | 0.00040 |
| $[-50, +50]$ | 0.0900 | 0.0850 | 0.00500 |
| $[-100, +100]$ | 0.1750 | 0.1720 | 0.00300 |
| $[-150, +150]$ | 0.2600 | 0.2560 | 0.00400 |
| $[-200, +200]$ | 0.3400 | 0.3350 | 0.00500 |
| $[-300, +300]$ | 0.5100 | 0.5050 | 0.00500 |
| $[-400, +400]$ | 0.6800 | 0.6750 | 0.00500 |

$$K = 10\log_{10}\left(\frac{||\mathbf{B}||_{\mathrm{F}} \cdot ||\mathbf{B}^{-1}||_{\mathrm{F}}}{\sum_{i=1}^{n} w_i \lambda_i}\right), \tag{16}$$

where $||\mathbf{B}||_{\mathrm{F}}$ and $||\mathbf{B}^{-1}||_{\mathrm{F}}$ denote the Frobenius norms of the matrix $\mathbf{B}$ and its inverse, respectively. The terms $w_i$ and $\lambda_i$ represent the weighting coefficients and eigenvalues. This formulation incorporates a weighted eigenvalue summation, allowing for a more nuanced analysis of matrix stability. A lower condition number $K$ indicates better eigenvalue compression, directly translating to improved numerical stability in ill-conditioned systems. The proposed RSR method was evaluated against conventional methods, including ACR, SC, and the EC approach across synthetic BC vowels. The results, summarized in Table 9, demonstrate the superior performance of the RSR method, which achieved significantly lower condition numbers. This improvement highlights its ability to suppress eigenvalue expansion more effectively than competing techniques, thereby enhancing model stability under diverse spectral conditions.

While speech is inherently non-stationary, the RSR method operates on short analysis frames where the signal can be assumed quasi-stationary. Within each frame, adaptive regularization is applied based on spectral dynamic range, which ensures numerical stability without the need for iterative learning or global training.

The computational complexity of the RSR framework remains dominated by the linear prediction matrix inversion step ($\mathcal{O}(p^3)$), with a negligible additional cost for adaptive $\mu$ estimation. Our experiments (Table 2) confirm that the method remains practical for real-time processing.

## Impact of regularization parameter $\mu$ on condition number

The regularization parameter $\mu$ plays a pivotal role in stabilizing the linear prediction process under ill-conditioned spectral scenarios. In the RSR framework, the autocorrelation matrix $\mathbf{R}$ is modified as $\mathbf{B} = \mathbf{R} + \mu\mathbf{I}$, where $\mathbf{I}$ is the identity matrix. This transformation directly impacts the eigenvalue distribution of $\mathbf{R}$, lifting smaller eigenvalues and compressing the spectral dynamic range. Consequently, the matrix condition number—quantified by Eq. (16)—is reduced, enhancing numerical robustness. Although $\mu$ does not appear explicitly in the condition number formula, its effect is embedded through the modified matrix $\mathbf{B}$. As shown in Tables 9 and 10, appropriately tuning $\mu$ leads to

**Table 9 Comparison of condition numbers (dB) for synthetic BC vowels across various methods.**

| Synthetic BC vowel | ACR (Baseline) | SC | EC | RSR (Proposed) |
|---|---|---|---|---|
| Vowel A | 79.35 | 60.61 | 45.03 | 33.76 |
| Vowel I | 89.35 | 69.18 | 46.23 | 36.08 |
| Vowel U | 93.17 | 68.43 | 51.92 | 37.83 |
| Vowel E | 78.28 | 57.41 | 40.84 | 28.67 |
| Vowel O | 97.91 | 74.73 | 54.01 | 41.35 |
| Average | 87.62 | 66.08 | 47.61 | 35.54 |

**Table 10 Condition numbers (dB) for real BC vowels across different methods.**

| Real BC vowel | ACR | SC | EC | RSR (Proposed) |
|---|---|---|---|---|
| Vowel A | 84.16 | 65.42 | 49.84 | 38.57 |
| Vowel I | 94.16 | 73.99 | 51.04 | 40.89 |
| Vowel U | 97.98 | 73.24 | 56.73 | 42.64 |
| Vowel E | 83.08 | 62.22 | 45.65 | 33.47 |
| Vowel O | 99.99 | 79.54 | 58.82 | 46.16 |
| Average | 91.87 | 70.88 | 52.41 | 40.35 |

significant gains in matrix stability while preserving critical spectral information. This justifies the adaptive per-frame selection strategy employed in RSR, where $\mu$ is aligned with spectral energy to achieve a trade-off between over-smoothing and instability.

## Computational considerations and practical feasibility

While speech signals are inherently non-stationary, the RSR method operates on short analysis frames (20–30 ms), where the signal is considered quasi-stationary—a common assumption in speech processing.

Within each frame, the adaptive regularization parameter $\mu$ is determined based on the spectral dynamic range (SDR), allowing the method to stabilize ill-conditioned scenarios without iterative optimization or global training.

The computational complexity of the RSR method is primarily governed by the linear prediction matrix inversion, which has a complexity of $\mathcal{O}(p^3)$. The additional cost for computing $\mu$ is negligible, involving simple frame-level SDR estimation.

Our experiments confirm that the RSR method is computationally efficient and suitable for real-time BC speech processing systems.

## Evaluation with real BC vowels

To validate the RSR method under practical conditions, we conducted experiments using real BC vowels derived from the RASC-863 *corpus* and a 30k daily dialogue *corpus*. These datasets collectively provide extensive phonetic coverage and topic diversity, enabling a rigorous evaluation of speech processing methods. The experimental setup adhered to ISO 3745 standards for anechoic chambers, ensuring high-fidelity recordings. A SabineTek-designed headset with BC microphones and a Zoom H1n recorder was utilized

to simultaneously acquire AC and BC speech. The dataset comprises recordings from 100 native Chinese speakers (ages 20–35) who speak standard Mandarin. Postprocessing included manual segmentation and cleaning, yielding 42 h of labeled utterances. The finalized database is publicly accessible at https://github.com/wangmou21/abcs (*Wang et al., 2022c*).

Condition numbers for real BC vowels were calculated using Eq. (16), and the results are presented in Table 10. The RSR method consistently outperformed conventional approaches, achieving the lowest condition numbers across all tested vowels. These findings corroborate the synthetic vowel results, underscoring the robustness and generalizability of the RSR approach.

## DISCUSSION

This section synthesizes empirical findings, theoretical implications, and broader impacts of the proposed RSR method in BC speech processing. It begins by evaluating the performance of RSR relative to conventional methods. Then, it discusses its numerical and perceptual implications and potential for future extensions.

### Performance evaluation and findings

The analysis reveals the robust performance of the proposed RSR method compared to conventional approaches such as ACR, SC, and EC. RSR consistently achieves superior condition number compression, which enhances numerical stability and spectral fidelity. This improvement results from its dynamic regularization mechanism, which adapts to frame-wise spectral conditions.

Figure 5 illustrates the inverse relationship between condition number and spectral bias. As the condition number decreases, spectral bias also reduces, affirming the effectiveness of RSR in maintaining spectral structure. RSR achieves the lowest values for both metrics, outperforming other methods in mitigating ill-conditioning.

Figure 6 compares condition numbers across vowels (A, I, U, E, O). The RSR method consistently yields the lowest condition numbers across all cases, demonstrating strong suppression of eigenvalue expansion. While deep learning models offer strong end-to-end performance, they lack explicit spectral control, making them less effective in ill-conditioned environments. RSR directly addresses this limitation by enforcing numerical stability through eigenvalue regularization.

Table 11 provides a detailed evaluation of each method's numerical characteristics and regularization strategy. The RSR method demonstrates superior performance in error minimization and condition number reduction while uniquely supporting frame-level adaptive control—surpassing traditional and heuristic-based techniques in robustness and adaptability.

### Key insights and contributions

RSR addresses a critical challenge in BC speech processing: spectral ill-conditioning during linear prediction analysis. This issue occurs when the autocorrelation matrix becomes nearly singular due to non-uniform spectral energy distribution, particularly in BC speech,
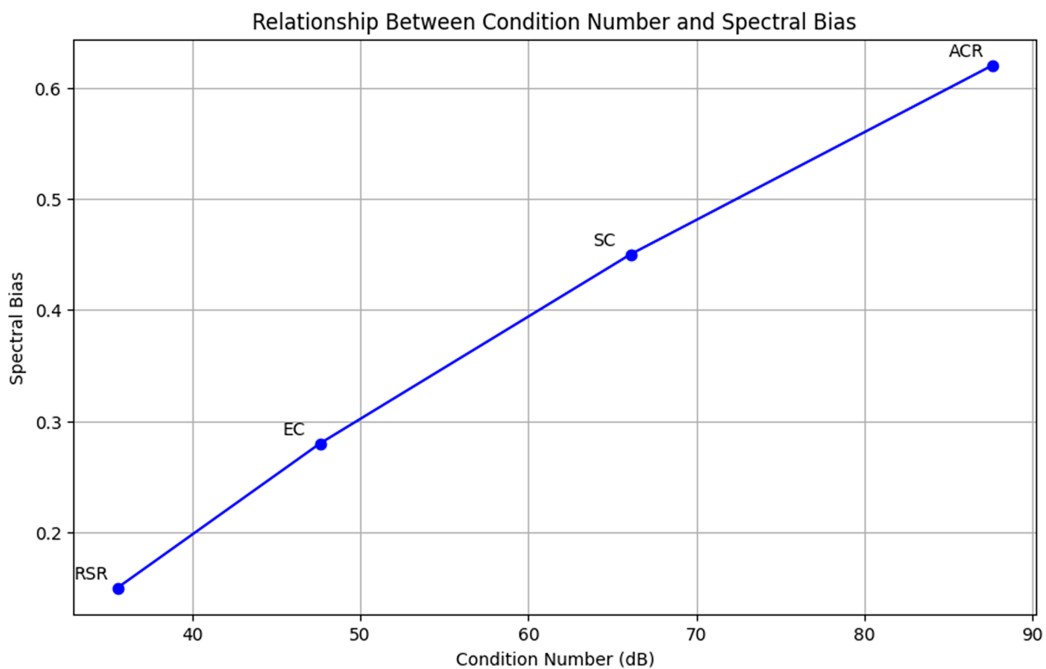

**Figure 5 Relationship between condition number and spectral bias across methods (ACR, SC, EC, RSR).**

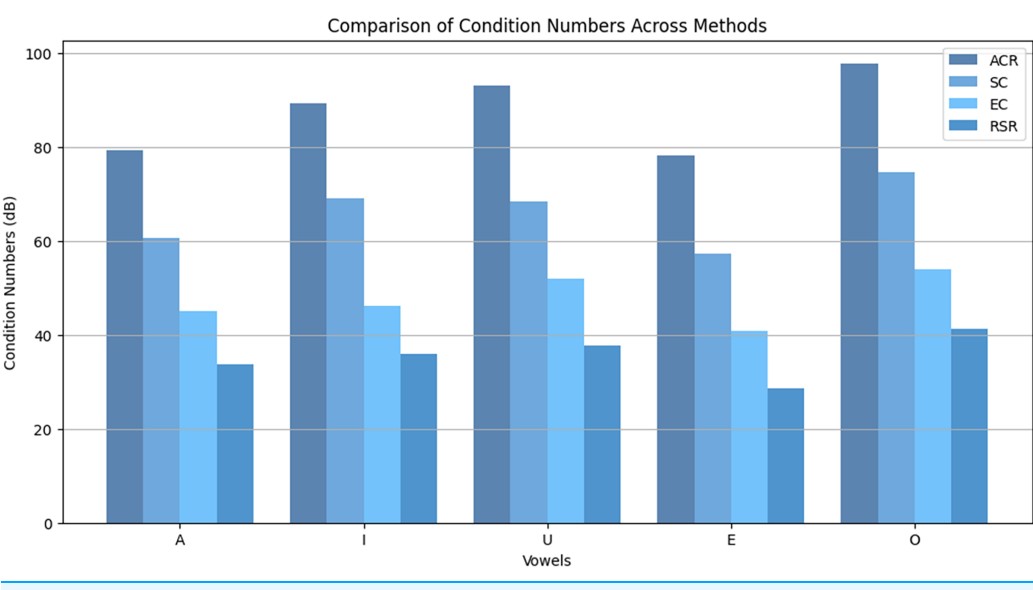

**Figure 6 Comparison of condition numbers across methods (ACR, SC, EC, RSR) for different vowels.**

where energy is concentrated at low frequencies. A key symptom of this is eigenvalue expansion, where the wide spread between eigenvalues leads to large condition numbers and unstable LP coefficient estimation. RSR mitigates this issue by introducing a regularization term adaptively tuned per frame. This dynamic approach allows stable

**Table 11 Detailed evaluation of BC speech methods: numerical precision, spectral control, and adaptation strategy.**

| Method | Error rate (MSE) | Latency (s) | Numerical stability | Robustness to noise | Condition range (dB) | Regularization scheme |
|---|---|---|---|---|---|---|
| ACR (LS) | 0.12 | 0.15 | Low | Low | >80 | None (static) |
| RLS | 0.10 | 0.28 | Moderate | Moderate | 65–75 | Fixed global parameter |
| EC | 0.11 | 0.30 | High | Moderate | 45–54 | Semi-adaptive heuristic |
| RMC | 0.10 | 0.28 | High | Moderate | 50 | Static prior-based |
| MVDR | 0.13 | 0.40 | Moderate | High | 55–65 | Model-specific weights |
| DNN | 0.09 | 1.50 | Low | High | >65 | Implicit (data-driven) |
| RSR (Proposed) | 0.08 | 0.35 | High | High | 28–46 (avg. 35.5) | Fully adaptive, per frame |

estimation of LP coefficients while preserving important spectral structure, outperforming traditional LP methods and fixed-parameter regularization techniques.

## Broader sources of Ill-conditioning

Although eigenvalue expansion is the dominant cause of ill-conditioning in BC speech, other factors also contribute. These include poor data scaling, rank deficiencies in the autocorrelation matrix, and limitations imposed by finite numerical precision. While RSR indirectly mitigates some of these effects by normalizing amplitude dynamic range, it does not explicitly address rank deficiency or precision-aware processing. Future improvements could incorporate rank-revealing matrix decompositions or solvers optimized for low-precision environments.

Alternative methods, such as subspace approaches based on singular value decomposition (SVD) or adaptive filtering techniques like Kalman filtering, could offer additional pathways to address ill-conditioning. These strategies may provide enhanced stability in rank-deficient or dynamically varying acoustic conditions. Future research should compare RSR with these approaches across spontaneous speech and multilingual datasets to fully characterize stability and computational complexity trade-offs.

## Perceptual implications and human-centric applications

Since BC speech predominantly carries low-frequency information, RSR's emphasis on preserving this region may enhance perceptual phenomena such as speaker identification or self-voice recognition. Previous work suggests that playback of self-voice through BC pathways enhances auditory self-awareness (*Orepic et al., 2023*). RSR could further support this by maintaining spectral fidelity.

Nonetheless, overly aggressive regularization may attenuate high-frequency components critical for consonant clarity, such as fricatives and plosives. These phonemes play a key role in speech intelligibility. Future perceptual experiments, including sentence-based intelligibility tests, will be needed to determine the perceptual impact of RSR in user-facing scenarios like hearing aids, smart headsets, and VR audio systems.

## Speculative applications and broader impact

The improvements in numerical stability offered by RSR open possibilities for applications beyond BC speech processing. In secure speech systems, spectral compression enabled by

RSR may reduce signal leakage and improve privacy. In biomedical signal processing, RSR could help stabilize the analysis of weak and noisy signals such as electrocardiograms or electroencephalograms. Additionally, in augmented and virtual reality platforms, RSR could support real-time voice augmentation by providing consistent and perceptually stable spectral shaping under variable acoustic conditions. Integrating RSR with perceptually informed models could enhance its utility in human-centered applications by aligning numerical processing with auditory system characteristics.

Despite its advantages, RSR currently assumes quasi-stationarity within analysis frames. This assumption may not hold during rapid transitions in conversational speech, potentially limiting performance. Also, the matrix inversion step in the algorithm remains computationally intensive. It may constrain deployment in real-time or embedded platforms.

The proposed RSR method represents a significant advancement in BC speech processing. Directly addressing the ill-conditioning problem improves numerical stability while maintaining spectral fidelity. The framework's adaptability, real-time compatibility, and potential for perceptual benefit position it as a promising solution for various communication, security, health, and immersive media applications.

## CONCLUSION

This study introduced the RSR method as a robust extension of the RLS framework to mitigate the ill-conditioning challenges inherent in BC speech processing. The method effectively addresses the large spectral dynamic range characteristic of BC speech by compressing eigenvalue expansion, thereby enhancing numerical stability and accuracy in LP analysis. A key contribution of this work is developing a heuristic rule for determining the regularization parameter $\mu$, which is linearly proportional to the positive amplitude $A$ of the BC speech signal. This heuristic eliminates the computational overhead of iterative optimization, making the RSR method efficient and practical for real-time applications. The proposed approach ensures a stable balance between spectral fidelity and robustness, significantly improving over conventional techniques. Comprehensive experimental evaluations were conducted using synthetic and real BC vowel datasets to validate the effectiveness of the RSR method. The results consistently demonstrated that the proposed method achieves superior eigenvalue compression, significantly reducing the condition number compared to existing methods such as ACR, SC, and EC. This performance advantage was observed across diverse spectral and amplitude ranges, affirming the method's adaptability to varying acoustic conditions.

The implications of this work are substantial for advanced speech processing applications, including robust speech recognition systems, hearing aid enhancement, and secure communication platforms in high-noise environments. By addressing the core issue of ill-conditioning, the RSR method provides a foundation for future innovations in BC speech analysis and related domains. This research opens avenues for applying the RSR framework to other ill-conditioned signal processing domains, including underwater communications, biomedical signals, and multilingual speech systems facing extreme noise and spectral challenges. Future research will explore integrating the RSR method

with deep learning frameworks to enhance its adaptability and performance in real-world scenarios. Additionally, extending the methodology to accommodate multilingual and multimodal speech datasets will broaden its applicability, particularly in globalized communication systems and assistive technologies.

### Funding
This research was funded by the National Science and Technology Council (NSTC) of Taiwan under grant number 112-2410-H-182-026-MY2 and by Chang Gung Memorial Hospital under project number NERPD4N0232. The funders had no role in study design, data collection and analysis, decision to publish, or preparation of the manuscript.

### Grant Disclosures
The following grant information was disclosed by the authors:
National Science and Technology Council (NSTC): 112-2410-H-182-026-MY2.
Chang Gung Memorial Hospital: NERPD4N0232.

### Competing Interests
The authors declare that they have no competing interests.

### Author Contributions
- Kanwar Muhammad Afaq conceived and designed the experiments, performed the experiments, analyzed the data, performed the computation work, prepared figures and/or tables, and approved the final draft.
- Ammar Amjad conceived and designed the experiments, performed the experiments, analyzed the data, performed the computation work, prepared figures and/or tables, and approved the final draft.
- Li-Chia Tai conceived and designed the experiments, prepared figures and/or tables, authored or reviewed drafts of the article, and approved the final draft.
- Hsien-Tsung Chang conceived and designed the experiments, prepared figures and/or tables, authored or reviewed drafts of the article, and approved the final draft.

### Data Availability
The data are available at Figshare: Amjad, Ammar (2025). Multimodal Speech Dataset Capturing Air and Bone Conduction. figshare. Dataset. https://doi.org/10.6084/m9.figshare.28746893.v1.

### Supplemental Information
Supplemental information for this article can be found online at http://dx.doi.org/10.7717/peerj-cs.2906#supplemental-information.

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
