# Peer review of "Adaptive regularized spectral reduction for stabilizing ill-conditioned bone-conducted speech signals"

_PeerJ Computer Science, doi:10.7717/peerj-cs.2906_

## Round 0.1 · original submission · Major Revisions

· Academic Editor

Major Revisions

Please respond to the comments from all 3 reviewers

Reviewer 1 ·

Basic reporting

All comments have been added in detail to the last section.

Experimental design

All comments have been added in detail to the last section.

Validity of the findings

All comments have been added in detail to the last section.

Additional comments

Review Report for PeerJ Computer Science
(Mitigating ill-conditioning in bone-conducted voice signals: a novel regularized spectral compression method)

1. Within the scope of the study, a regularized spectral reduction method based on the regularized least squares framework is proposed in relation to linear prediction analysis.

2. In the introduction section, the content and importance of the subject, bone-conducted and air-conducted speech characteristics, and the contributions of the study are mentioned. The contributions section of the study needs to be detailed in a more emphatic way and in a way that will clearly indicate the originality point.

3. In the Related works section, the Enhanced Covariance method and its comparison with traditional linear prediction are mentioned. Although a certain level of literature is mentioned in this section, it is recommended that the comparison table-2 is prepared in more depth.

4. When the algorithm for the Enhanced Regularized Spectral Reduction method proposed in the study is examined and when the comparison is made in terms of situations such as Error Rate and Computational Time, it is observed that the study has a certain level of originality in terms of method.

5. In the Experiment section, it is understood that the shared feature types and their corresponding values are sufficient and appropriate for this study since they are compared with the literature. In addition, the results and performance evaluation sections given with the tables are sufficient. Increase and detail the comparisons in Table-11.

As a result, the study can contribute to the literature with the methodology it proposes, but attention should be paid to the above sections.

Reviewer 2 ·

Basic reporting

The authors have tried to improve the ill-condition of BC speech signal by using the regularized spectral reduction (RSR) method. The linearity of the regularization parameter is also investigated, and the performance of the RSR method has been compared with other methods by using condition numbers. The paper is well put together, and it has good novelty. However, there are some issues with this article and after correction of the below-mentioned points, the paper can be accepted.
1. What is the significance of the setting of the regularization parameter in the performance of the
RSR method for BC speech signals?

2. Adding regularized parameters means adding some noise to the conventional method. Is there any effect to the Least square method?

3. I suggest using a proper reference of Table 5 that supports generating synthetic AC speech through a set of LP coefficients.

Experimental design

4. I noticed some comparative analysis. If possible compare your work with [I] Ill-condition Enhancement for BC Speech Using RMC Method, https://doi.org/10.1007/s10772-024-10159-9; and [ii] Regularized Modified Covariance Method for Spectral Analysis of Bone-Conducted Speech; https://doi.org/10.2299/jsp.28.77

5. In Table 8, the SDR ranges from -100 to +100; if it is -50 to +100, how do you derive the regularized parameter?

6. For the Synthetic AC generation where the gain factor is K0 =0.1106 for 5 vowels. How do you validate it because 5 vowels do not represent the same fundamental frequency (Fo)? Explain with proper references. I suggest for each vowel, use different gain factors. Also, show the proper equation for gain factor calculation.

7. For ψ and φ are the filter coefficients set to 0.82 and 0.32, respectively, and the 20 dB attenuation for synthetic BC speech. But in Figure 3, there is no relation for 20dB degradation. You can check here in Figure. 2 https://doi.org/10.2299/jsp.28.77

Validity of the findings

8. Finding µ where k is 0.00170 which is very close to the k value of the RMC method https://doi.org/10.2299/jsp.28.77 where k is 0.00165. How do you claim this is a Novel RSR method? I suggest the title should be changed because this is never a novel method.

9. Regularized parameter optimization is the concept of deep learning and
machine learning. But how can you use in statistical approach where speech signal is non-stationary? Is there any time complexity issues arise? Explain properly.

Additional comments

10. Check all the references are properly cited.

Annotated reviews are not available for download in order to protect the identity of reviewers who chose to remain anonymous.

Reviewer 3 ·

Basic reporting

The manuscript is generally well-written, though some technical terms could be clarified for readers less familiar with LP methods.

While the authors cite foundational works on LP and BC signal processing, the manuscript could benefit from a larger discussion of alternative methods. There is an almost exclusive emphasis on one method in “Related works” section.

The discussion of the results is largely missing, and the discussion section mainly contains results.

Figures could be improved with clearer labels and captions.

Experimental design

The experimental setup for testing the RSR method on synthetic and real BC speech datasets is well-conceived, but additional details on the criteria used to fine-tune the regularization parameter would enhance reproducibility.

It would be useful to illustrate some of the properties of the real BC dataset, particularly regarding signal diversity and noise levels.

Validity of the findings

The authors effectively demonstrate eigenvalue compression using RSR.

The claim of improved numerical stability is reasonable, but a more thorough discussion on how spectral fidelity is maintained alongside stability would be beneficial.

While the proposed improvements in LP analysis show promise for the mentioned BC speech applications, such as communication in high-noise environments, it would also be helpful to discuss the validity in relation to human perception of such BC speech.

Additional comments

Line 57: It would be useful to briefly say what ill-conditioning is and expand on previous approaches.

Table 1: I suggest replacing this table with a table giving an overview of previous solution & their pros and cons. What you have in Table 1 is already present in the text and it is not so crucial to highlight in such a way for this paper

Equations 1 and 2: It might be useful to show LP and RSR also in matrix formats, since you are putting emphasis on eigenvalues. The relationship might not be obvious for all readers

80: The entire “Related Works” section is describing one alternative “work” - Enhanced Covariance (EC), setting up the stage for RSR. Are there any alternative approaches/methods that advanced LP compared to traditional ones?

88: “Eigenvalue expansion, ...” I would place this sentence in the previous section to increase clarity, and give 2-3 more sentences of general problem description

101-102: What do you consider by low and mid-range?

Algorithm 1: How are missing values handled? Could data imputation strategy play a role in your approach?

Table 3: How are the stability and noise tolerance measured?

Figure 1: Inconsistent use of “RSR” and “RLS” abbreviations. In the text, they are called. “Regularized Spectral Reduction” and “Regularized Least Square”. It is strange to have “e.g.” in y axis. You should indicate exactly what y axis represents. The same for the x axis – how do you manipulate noise level?

Figure 2: This Figure is completely unclear. What do the individual blocks and colors represent?

186: Use of words such as “rigorously” is subjective, especially when describing own work. This is present on several places throughout the manuscript

213: why these specific values (0.82 and 0.32)?

Figure 4: Mention that this is for /a/ vowel. Do other vowels look similar?

Equation 16: How is mu related to this formula?

Tables 6 and 7: It would be useful to show these tables as plots, with continuous mu values on x axis, to illustrate the behavior of mu better, especially in the vicinity of the optimum

283: Most of the discussion section is still results. The discussion is largely missing in the article.

Some suggested discussion points:

The manuscript discusses eigenvalue expansion as a primary contributor to ill-conditioning in BC speech analysis and emphasizes the benefits of the proposed RSR method in addressing this issue. However, ill-conditioning can also arise from other factors, such as data scaling, rank deficiency, or numerical precision issues in matrix computations. I suggest that the authors briefly discuss some additional potential sources and stress whether the RSR method inherently addresses any of these factors beyond eigenvalue spread, or if future enhancements are needed to tackle them.

Have the authors considered comparisons with other regularization techniques, or subspace methods like SVD or adaptive filtering approaches like Kalman filtering?

Application for human perception: if your approach is effectively changing the BC spectrum, which perceptual attributes do authors think it will affect – e.g. for speech and speaker intelligibility?

Speech: Considering that vowels have formant peaks in low frequencies that are more conserved in BC, how do you think your results would generalize to consonants or full sentences? Would any information be lost due to regularization that could lead to loss of intelligibility, e.g. of fricatives plosives?

Speaker: It is a well-known fact that hearing our voice in recordings does not sound natural to us, due to the lack of BC that is inevitably present while we speak. A recent study showed that playing our voice through BC increases self-voice recognition, which does not generalize to recognition of other voices (Orepic et al., 2023, RSOS). Would such perceptual advantages be present with your approach?

If anything, these examples could enrich the article with more speculative future applications, increasing the impact of this work.

Table 11: “improved”, “Better”, “higher” – compared to what exactly?

---

## Round 0.2 · accepted · Accept

· Academic Editor

Accept

Both reviewers agree you addressed their comments. If possible, I would suggest taking into account these final, minor suggestions.

Reviewer 1 ·

Basic reporting

All comments have been added in detail to the last section.

Experimental design

All comments have been added in detail to the last section.

Validity of the findings

All comments have been added in detail to the last section.

Additional comments

Review Report for PeerJ Computer Science
(Adaptive regularized spectral reduction for stabilizing ill-conditioned bone-conducted speech signals)

Thank you for the revision. Both the responses to the reviewer comments and the relevant changes to the paper are acceptable. Best regards.

Reviewer 3 ·

Basic reporting

The authors have successfully addressed and implemented all my comments. I have a few very minor suggestions below.

Table 1: it is not clear what the abbreviations stand for because the table comes before the text describing these approaches. I suggest placing Table 1 below the paragraph where you introduce other methods (line 103). Rows 50-60 that introduce RMC methods could be also placed after the row 103, to increase the flow. Column 2 “strengths wrt ACR” sounds strange if you have a row “ACR”, because it is then strength wrt itself. It could just be “Strengths”

Figure 1: I do not see the updated figure in the revised document, but the original one
Figure 2: what do green and yellow colors represent?

Experimental design

indicated above

Validity of the findings

indicated above

Additional comments

indicated above